# Examining Factors of Health Information Communicant Activeness of Chinese Residents in Outdoor Activities during Public Health Emergencies

**DOI:** 10.3390/ijerph20010838

**Published:** 2023-01-02

**Authors:** Jie Liu, Jinhong Zhang, Norliana Binti Hashim, Sharifah Sofiah Syed Zainudin, Siti Aishah Binti Hj Mohammad Razi

**Affiliations:** 1Faculty of Modern Languages and Communication, Universitiy Putra Malaysia, Serdang 43400, Malaysia; 2Faculty of Educational Sciences, Shanxi University, Taiyuan 030006, China

**Keywords:** public health emergencies, health information, perceived risk, situational theory of problem-solving

## Abstract

This study explores the influencing factors of residents’ outdoor health information communicant activeness under public health emergencies and analyzes the relationship between cognition, motivation, and dissemination behavior. Based on the theory of perceived risk and the Situational Theory of Problem-Solving (STOPS), this study builds a model demonstrating the factors that affect the health information communicant activeness of residents’ outdoor activities under public health emergencies and conducts empirical analysis through questionnaires and a structural equation model. Results showed that (1) perceived risk, problem recognition, and involvement recognition positively affected health information communicant activeness through situational motivation; (2) constraint recognition negatively affects health information communicant activeness through situational motivation; and (3) the referent criterion has a direct effect on communicant activeness. This study has great significance for understanding residents’ health information communicant activeness in outdoor activities and providing them with high-quality health information services.

## 1. Introduction

“Emergency” refers to natural disasters, accident disasters, public health events, and social security events that occur suddenly and cause or may cause serious social harm and require emergency response measures [1]. After the outbreak of SARS in 2003, the term “public health emergencies” (PHE) formally entered people’s minds and received extensive attention. Especially after the outbreak of COVID-19 in 2020, public health emergencies caused extensive discussion and reflection in society because of their wide range and threat to public health, and the impact on residents’ outdoor activities and the tourism industry was the most serious. With current news and information technology developments, health information communication plays an increasingly important role in promoting the exchange and sharing of information among residents during outdoor activities. From the existing research, the influence mechanism on health information communicant activeness of residents’ outdoor activities in public health emergencies is still vague, and the factors and how to influence are still unclear. Previous studies found that the factors affecting health information communicant activeness are mainly related to information content, information type, information quality, and information source [2,3].

Kyungsuk et al. [2] found that information quality and source reliability are key factors affecting information seeking and adoption. Westerman et al. [3] further investigated how available information affects people’s perception of source credibility. Hou et al. [4] compared the diffusion paths of different types of health information and put forward constructive suggestions for health information management. The discussion based on the characteristics of the information itself is a critical reference perspective for the study of the information communicant activeness of residents’ outdoor activities. However, the current research on the other perspective of health information communicant activeness - the perspective of residents themselves is less. The role of residents’ outdoor activities is vague in PHE. They can freely participate in all aspects of information exchange and have the ability to release, transmit, and even hinder the dissemination of information. Some studies believe that residents’ outdoor activities are more based on self-cognition, such as benefit perception [5] and perceived risk [5,6], and efficacy perception [6]. Therefore, the subjective will of residents is worthy of attention in the study of health information communicant activeness in PHE. Simply considering the information characteristics is not enough to restore the overall picture of the process of health information communicant activeness.

## 2. Literature Review

Existing scholars have mainly explored the motivation and willingness of social media users to participate in information dissemination from the users’ perspective, while there is little research on the health information communicant activeness of residents in outdoor activities at PHE. Sheldon et al. [7] pointed out that the motivation for using Instagram is mainly to meet the users’ needs for self-improvement, social interaction, entertainment, recording, and creation. Jiang and Yan [8] studied the influencing factors of social media communication based on the stimulus-organism response (S-O-R) theory and believed that perceived usefulness and perceived trust were the direct factors affecting communication behavior. These studies are mainly aimed at the daily information dissemination of users, with obvious social and entertainment motives. However, the applicability of this highly sensitive health information communicant activeness needs further discussion. Health information involves users’ privacy and physical and mental health and is sensitive information [9]. Users often participate in information dissemination activities to solve health problems after they perceive the disease risk, rather than for social interaction or entertainment. The perceived risk attitude framework [6] believes that disease risk is an important reason to promote the protection of individual information-seeking behavior. The cognitive theory believes that external stimuli (such as disease) generate the user’s information demand. The problem-solving situation theory further believes that in addition to problem cognition (demand cognition), the user’s motivation and behavior are also affected by the degree of involvement and the influence of self-efficacy [10]. Therefore, residents’ unique situation of “health problems” should be fully considered when studying the health information communicant activeness of outdoor activities of residents in PHE. Because of this, this paper integrates perceived risk and Situational Theory of Problem-Solving (STOPS) to study the health information transmission mechanism of residents’ outdoor activities in public health emergencies and analyzes the relationship between users’ psychological cognition and communication behavior in the context of health problems to further understand residents’ health communicant activeness further.

## 3. Theoretical Framework and Hypothesis

### 3.1. Perceived Risk

Perceived risk is an individual’s cognition, attitude, and judgment of various objective risks in the outside world. Health perceived risk is people’s cognition of various factors, activities, and common diseases that affect physical and mental safety and health [11]. Perceived risk focuses on subjective feelings and cognition, which can better explain the irrational behaviors caused by individual cognitive bias and effectively predict health problem decision-making and health behavior intervention. According to Cunningham’s [12] two-factor model, risk can be considered from the two dimensions of loss uncertainty and result harmfulness, thus determining the degree of individual perceived risk. When individuals perceive potential health risks, they tend to fall into negative emotions such as worry, anxiety, and even fear. Rosenboim et al. [13] found that fear has a significant impact on perceived risk; Zhang [14] believes that reasonable fear can arouse the attention of individuals to health risks and improve the audience’s compliance. Brewer et al. [15] demonstrated that perceived risk positively relates to health behavior. The protection motivation theory explains this phenomenon: individuals actively seek information and solutions under the influence of high-perceived risk to change their uneasy state out of self-protection. Perceived risk is often used as a predictor of protective behavior. Rimal and Hee [16] studied the relationship between perceived risk and protective attitudes toward breast cancer. Therefore, this study regards perceived risk as a factor that affects residents’ health and situational motivation. When the public perceives health risks, they tend to seek information to avoid them. The hypothesis was proposed as follows:

**H1:** *The residents’ perceived risk of health problems in PHE positively affects the situational motivation of health information behaviors*.

### 3.2. STOPS

Situation refers to the specific environment and state of cognitive factors in the information search and sharing process, which is the critical thinking base of the cognitive paradigm of information science [17]. According to the SOR model, when users perceive the disease risk and the information or knowledge, they will fall into a state of lack of information or knowledge [18]. The information or knowledge missing in this part of solving problems is the information demand. Due to the lack of knowledge, the cognition is insufficient and uncoordinated, resulting in the cognitive “broken belt,” which forms the concept of “problem” in this paper. The STOPS describes the relationship between the influencing factors, motivation, and information behavior in solving this “problem” in health; the perceived health risk is regarded as a problem-driven event. Under the stimulation of this situation, it recognizes health information or a knowledge gap (problem cognition) and then generates the motivation and behavior of health information dissemination. The whole health information communicant activeness is the process of users solving “problems” in a specific situation. STOPS was proposed by Kim and Grunig [19] based on the Situational Theory of Publics. Compared with the Situational Theory of Publics, it downplays the “public situation” [20], regards human thinking and exploration as the process of problem-solving, and focuses on the antecedents of individual communication and information behavior in the process of problem-solving [21]. The theory includes four independent variables: problem recognition, involvement recognition, constraint recognition, and referent criterion. It takes situational motivation as the intermediary variable and six dependent variables, including information forefending, information permitting, information forwarding, information sharing, information seeking, and information processing. It fits the concept of situation and describes the process of an individual’s cognitive internalization to the activity level in information communication from three levels: cognition, motivation, and action. Kim and Grunig [19] suggested that the STOPS can be applied to the communication field outside the public relations field, such as political communication, climate communication, and food safety issues. Dai [22] used the STOPS model to explore the cognition and communication behavior of Chinese and American women’s cervical smear examination and found that this model is more suitable for the field of health communication. In addition, Li et al. [23] studied the sexual health information communicant activeness among migrant workers based on the STOPS, again proving the applicability of the model in the field of health communication. Therefore, based on the STOPS, this paper considers the health information communicant activeness of residents in PHE and discusses whether the model is also universal in the PHE environment in order to provide a reference for the health dissemination research in PHE.

Problem recognition refers to the extent to which people realize they have problems because they lack something that cannot be solved immediately. Residents aware of the disease risk realize that the health information in public health emergencies is beneficial, and they are in a state that lacks such information, which will positively impact their communication behavior. That conforms to the cognitive concept of information demand in the information acquisition behavior conceptual model of Wilson [24]. Therefore, this paper believes that problem recognition plays a role in promoting the motivation of communication behavior and puts forward the following hypothesis:

**H2:** *Residents’ problem recognition in PHE positively affects the situational motivation of health information behaviors*.

Constraint recognition is an obstacle that people are aware of when solving problems. This obstacle restricts their ability to solve problems [20]. Once people realize they cannot solve problems, they are likely to have a negative attitude and make no effort. The constraint on health information communicant activeness in PHE is mainly the issue of information authenticity. Users are limited by their ability to identify health information and cannot identify correct health information. This paper believes that constraint recognition has an obstacle effect on the user’s health information communicant activeness. Therefore, it puts forward the hypothesis:

**H3:** *Residents’ constraint recognition in PHE negatively affects the situational motivation of health information behaviors*.

Involvement recognition is people’s perception of an association between themselves and problems [20]. Research shows that involvement recognition will make the public different in learning mode, cognitive state, and information processing [25]. When the public finds a connection between themselves and the problem, it is very likely to take positive action to change the status quo. The higher the degree of association, the stronger the public’s willingness to change and the more likely they are to take positive communication actions. Therefore, the following hypothesis is proposed:

**H4:** *Residents’ involvement and recognition in PHE positively affect the situational motivation of health information behaviors*.

Referent criteria refer to the public’s experience and subjective judgment in solving problems. When residents have significant perceived risk and health problem awareness in outdoor activities and are aware that the problem has a strong relationship with themselves, and if they have enough knowledge and experience in disease prevention and treatment of public health events, they may have a strong motivation to search for relevant information. Therefore, this paper proposes the following hypothesis:

**H5:** *Residents’ referent criterion positively affects their communicant activeness in PHE*.

Problem recognition, involvement recognition, and constraint cognition need to use situational motivation as an intermediary variable in STOPS to finally reach the level of the behavior-dependent variable. In social psychology, motivation variables are efficient. Situational motivation is proposed as an intermediary variable between independent and dependent variables, defined as individuals’ intention to think about and change problems. When individuals have a strong desire to understand and change, they will take action, thus promoting communication activeness. This paper proposes the following hypothesis:

**H6:** *Residents’ situational motivation positively affects their communicant activeness in PHE*.

In addition, there are six communication behaviors in the problem-solving situation theory, which divides information selection, information transmission, and information acquisition into active and passive situations, including the communication behaviors that residents can take in PHE. Information selection includes information defense (active) and information acceptance (passive). Based on the above discussion, a conceptual model of the factors affecting health information communication activeness in PHE is constructed, as shown in Figure 1.

## 4. Methodology

The questionnaire in this study is mainly based on the maturity scale of related research and modified in combination with the actual survey, ensuring reliability, validity, and feasibility. The questionnaire is divided into two parts. The first part mainly collects the basic information of the respondents, and the second part sets the observation items around the previous hypotheses. The research adopts Likert’s seven-level scale method to collect quantitative data. The degree is from less to more, from negative to positive, and the score is 1–7. Specific measurement items and relevant variables are shown in Table 1. From May to August 2022, this survey was randomly distributed by “Wenjuan Star” in the form of a network, and a total of 600 responses were received. Through screening and excluding the questionnaires from abroad, with the same filling contents but too short a filling time, a total of 550 effective questionnaires were obtained, and the adequate questionnaire proportion was 91.67%. Generally, at least ten samples should measure each significant variable in the structural equation model [26]. Therefore, 500 effective samples from this study met the modeling requirements. A structural equation model was used to analyze the path relationship between variables in the questionnaire data, and SPSS and AMOS were used to complete the analysis process.

## 5. Data Analysis

### 5.1. Demographic Analysis

After data cleaning and pre-processing, the descriptive demographic statistics are presented in Table 2. The following information will explore the characteristics of the population of Chinese residents in terms of five essential aspects: gender, age, educational background, frequency of outdoor activities, and amount of travel purchased each time.

### 5.2. Reliability and Validity Analysis

The reliability and validity of the model should be verified first. Generally, the normalized load factor, Cronbach’s α, composite reliability (C.R.), and average variance extraction value (AVE) are used to test, as shown in Table 3. Cronbach’s α values of all variables in this study are greater than 0.7, indicating that the scale has high reliability. The questionnaire has high convergence validity when the AVE value is higher than 0.5, and the C.R. value is above 0.7. The AVE value of each variable in Table 3 is greater than 0.5, and the C.R. value is greater than 0.7. Therefore, the scale has high convergence validity.

Discrimination validity is mainly tested by calculating the relationship between the square root value of AVE and the correlation coefficient between variables, as shown in Table 4. It can be seen from the table that the correlation coefficients of all variables are less than the square root value of AVE, which indicates that the questionnaire has good discrimination validity. Therefore, the questionnaire is reasonable and has high validity.

### 5.3. Model Fitting Test

Amos 24.0 software was used for the model fitting test to complete the confirmatory factor analysis (CFA) of questionnaire data. The model fitting results are shown in Table 5. The indexes used in the model fitting test include absolute fitting indexes (the ratio of chi-square value and degree of freedom, GFI, and RMSEA), value-added fitting indicators (IFI and CFI), and comprehensive fitting indexes (PGFI and PNFI). The results in Table 5 show that the model can fit well.

### 5.4. Hypothesis Test

The maximum likelihood estimation method is used to estimate the model’s parameters. The test results are shown in Table 6, and the standardized path coefficient of the research hypothesis is shown in Figure 2. The path coefficient represents the standardized path coefficient, which reflects the degree of correlation between variables. “***” represents *p* < 0.001, indicating that the path is very significant; “**” represents *p* < 0.01, indicating that the path is relatively significant; and “*” represents *p* < 0.05, indicating that the path is significant. The data in Table 6 support the hypotheses H1, H2, H3, H4, H5, and H6 mentioned above.

## 6. Conclusions

Based on the theory of perceived risk and STOPS, this paper builds a theoretical model from the psychological perspectives of perceived risk, problem recognition, involvement recognition, and constraint recognition and studies the influencing factors of health information communicant activeness in PHE. Through the above data, the following findings can be obtained:

(1) Perceived risk positively affected health information communicant activeness through situational motivation in PHE. When residents are aware of their health risks, they will actively seek relevant information to solve their health problems, which is consistent with the perceived risk attitude framework and the research conclusion of Yang et al. [30];

(2) Problem recognition positively impacts health information communicant activeness through situational motivation, indicating that the more genuine the user’s lack of health information, the stronger the information demand generated and the stronger the idea of taking action. This conclusion is consistent with the research results of the intelligence demand theory and public communication behavior of Li et al. [29]. In addition, this study also found that problem recognition had the most significant impact on the residents’ health information communicant activeness through situational motivation;

(3) Constraint recognition negatively affects health information communicant activeness through situational motivation in PHE. This result is consistent with the previous study by Kim [19]. Constraint recognition discourages communication behaviors such as information seeking and attending, even if communicators have high problem recognition and perceived involvement [31]. Research by Grunig [32] in Colombia showed that people are less likely to communicate about “problems or issues about which they believe they can do little or about behaviors they do not believe they have the personal efficacy to execute” [33];

(4) Involvement recognition positively affects health information communicant activeness through situational motivation in PHE, indicating that when users participate in health communication, they first consider their relevance to this issue. The involvement theory [25] believes that people will make behavior decisions quickly and irrationally when they are highly involved. When people themselves or close people around them are troubled by health problems, the initiative of users to participate in health information exchange activities will be enhanced;

(5) In PHE, there is a significant positive relationship between health information, situational motivation for outdoor activities, and communicant activeness. The motivation theory holds that motivation represents the initiative of individuals to take action [34]. Situational motivation is a mediator variable generated by the three antecedents of perceived risk, problem recognition, and involvement recognition. The higher the residents’ awareness of health issues, involvement recognition, and perceived risk, the stronger the situational motivation, and the greater the possibility of adopting communicant activeness, which is consistent with the research result of Kim [19];

(6) In PHE, residents’ reference criteria positively affect health information communicant activeness, consistent with Kim [19]. If people have difficulty retrieving a workable solution from internal storage, they will likely show greater communicative action when composing a novel solution. At the same time, the problem holder will be eager to select and give information when a referent criterion is present. Overall, a more substantial subscription to a referent criterion will lead to higher communicative action in problem-solving. When a problem holder retrieves such a self-fulfilling or complacent referent (e.g., a goal, a desire, or a preference), this will strongly influence the interpretations and selection of the data encountered during problem-solving [35]. The more substantial presence of such self-fulfilling decisional referents will result in more information seeking, selection, and problem-solving.

## 7. Implications for Theories and Practitioners

From a theoretical perspective, this study has verified the adaptability of STOPS in the field of PHE and health communication in China for the first time. The results show that the two factors of problem recognition and involvement recognition of the original model are still key factors in the health information communicant activeness of residents’ outdoor activities in PHE, and the negative effect of constraint cognition is also confirmed. Secondly, considering the sensitivity and risk of health problems, the perceived risk theory and the STOPS are integrated, and the perceived risk is incorporated into the STOPS as a variable, thus developing the STOPS. The results show that risk perception is effective in the new model and is an essential factor influencing health information and situational motivation. In practice, this study helps health information communicators consider their communication motivation, behavior, and influencing factors in the PHE environment from the audience’s perspective, to help professional health service providers and health educators provide better health and education services. Furthermore, it improves public health literacy and the health information service system and promotes the implementation of a healthy Chinese strategy.

## 8. Limitations and Future Research

The results of this study have some limitations. First, the number of samples is limited, and there are only 550 valid questionnaires, which will affect the validity of the research results to a certain extent. Secondly, the population targeted by the questionnaire is mainly concentrated in fixed areas, which may lack certain representativeness compared to the residents of China. Third, the population under 20 years old should also be included in the survey. In addition, the Situational Theory of Publics uses three variables of problem recognition, participation recognition, and constraint recognition to divide people into non-public, potential public, knowing public, and strategic public. These four groups have different initiatives in information processing and importance in the information communication process. In this paper, there is no in-depth study on these four types of publics, but it is helpful to distinguish them in health education popularization, which is the direction of future research.

## Figures and Tables

**Figure 1 ijerph-20-00838-f001:**
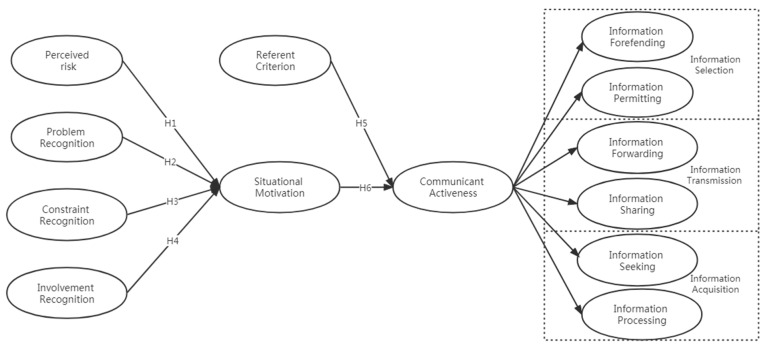
Research model diagram.

**Figure 2 ijerph-20-00838-f002:**
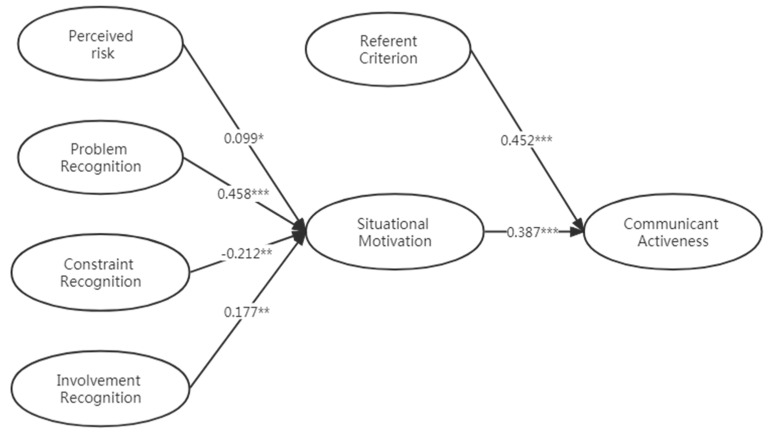
Normalized path coefficient diagram. * *p* < 0.05; ** *p* < 0.01; *** *p* < 0.001.

**Table 1 ijerph-20-00838-t001:** Measurements of variables.

Variables	Items	Literature Reference
Perceived risk	There is a likelihood of me contracting COVID-19 while traveling domestically.	Lan-Hsun [27]
	I am concerned that I would spread COVID-19 if I contracted it during domestic travel.	
	There is a likelihood of me contracting COVID-19 while eating outside.	
Problem recognition	I often get health information from outdoor activities.	Kim [19]
	The health information I learned during outdoor activities meets my needs.	
	I lack knowledge of health.	
	Health information during outdoor activities will affect many people.	
Involvement recognition	Health problems are closely related to my life.	Chen [25]
	Health problems affect people around me to a large extent.	
	My health problems have affected me to a great extent.	
	I pay more attention to health issues than the people around me.	
Constraint recognition	Health problems cannot be solved.	Deng [6]
	I cannot take action to deal with my health problems.	
	As long as I am willing, my efforts can solve health problems.	
	The health information in the process of outdoor activities is different. I know how to avoid health problems.	
Referent criterion	I already know enough about health.	Li et al. [23]
	In public health emergencies, I have my own opinions and choices.	
	I know how to prevent disease transmission in public health emergencies.	
	I can rely on experience to treat diseases in public health emergencies.	
Situational motivation	I want to learn more about health.	Zhang et al. [28], Li [29]
	I will spend more time thinking about health.	
	I think it is helpful to understand health information.	
Information selection	During outdoor activities, I can pick out correct and valuable health information.	Li [5], Deng [6]
	I am willing to accept all kinds of health information from different sources.	
Information transmission	I will actively transmit and share health information during outdoor activities.	
	I am willing to participate in the discussion on health.	
Information acquisition	During outdoor activities, I will actively seek health information.	
	I will pay attention to the health information found during outdoor activities.	

**Table 2 ijerph-20-00838-t002:** Demographic description.

Items	Categories	N	Percent (%)
Gender	Male	246	44.73%
	Female	304	55.27%
Age	20–25	72	13.09%
	26–30	288	52.36%
	31–35	134	24.36%
	36–40	56	10.18%
Educational background	High school and below	96	17.45%
	College	188	34.18%
	Graduate student and above	266	48.36%
Frequency of outdoor activities	Daily	34	6.18%
	Weekly	160	29.09%
	Monthly	356	64.73%
	Under RMB 500	28	5.09%
Amount of travel purchase	RMB 501–1000	178	32.36%
	RMB 1001–2000	224	40.73%
	RMB Above 2000	120	21.82%
Total		550	

**Table 3 ijerph-20-00838-t003:** Reliability and convergent validity analysis of scales.

Variables	Items	Cronbach’s α	AVE	C.R.
Perceived Risk	PRI1–PRI3	0.746	0.514	0.756
Problem Recognition	PRE1–PRE4	0.852	0.592	0.853
Constraint Recognition	CRE1–CRE4	0.833	0.559	0.835
Involvement Recognition	IRE1–IRE4	0.897	0.694	0.901
Referent Criterion	RCR1–RCR4	0.782	0.591	0.793
Situational Motivation	SM1–SM3	0.845	0.647	0.846
Communicant Activeness	CA1–CA6	0.899	0.642	0.914

**Table 4 ijerph-20-00838-t004:** Discrimination validity: Pearson correlation and AVE square root value.

	1	2	3	4	5	6	7
(1) Perceived Risk	0.717						
(2) Problem Recognition	0.381	0.769					
(3) Constraint Recognition	−0.361	−0.564	0.747				
(4) Involvement Recognition	0.400	0.345	−0.545	0.833			
(5) Referent Criterion	0.363	0.577	−0.524	0.435	0.700		
(6) Situational Motivation	0.400	0.598	−0.541	0.452	0.569	0.804	
(7) Communicant Activeness	0.394	0.616	−0.643	0.518	0.648	0.641	0.801

Note: The value on the diagonal is the AVE square root value.

**Table 5 ijerph-20-00838-t005:** Model fit index.

Index	Standard	Value
χ^2^/df	<5	3.622
GFI	>0.8	0.866
PGFI	>0.5	0.713
IFI	>0.9	0.909
CFI	>0.9	0.908
PNFI	>0.5	0.776
RMSEA	<0.08	0.069

**Table 6 ijerph-20-00838-t006:** Path coefficient of the research model.

Hypothesis	Path Hypothesis	Path Coefficient	*SE*	*z* (C.R.)	*p*-Value	Test Results
H1	Perceived risk → Situational Motivation	0.099	0.034	2.040	0.041 *	Supported
H2	Problem Recognition → Situational Motivation	0.458	0.067	7.318	0.000 ***	Supported
H3	Constraint Recognition → Situational Motivation	−0.212	0.065	−3.050	0.002 **	Supported
H4	Involvement Recognition → Situational Motivation	0.177	0.036	3.320	0.001 **	Supported
H5	Referent Criterion → Communicant Activeness	0.452	0.048	8.210	0.000 ***	Supported
H6	Situational Motivation → Communicant Activeness	0.387	0.045	7.554	0.000 ***	Supported

* *p* < 0.05; ** *p* < 0.01; *** *p* < 0.001.

## Data Availability

Everyone who concerns about this topic and can obtain the data generated during the study from gs58630@student.upm.edu.my.

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
