# Peer review of "Examining Factors of Health Information Communicant Activeness of Chinese Residents in Outdoor Activities during Public Health Emergencies"

_ijerph, 2023, doi:10.3390/ijerph20010838_

Round 1

Reviewer 1 Report

The information flow during the recent pandemic created a very controversial problem space, and the experience has shown that even the most intuitive hypotheses must be meticulously studied and confirmed via proper data collection and analysis. In this light, this is an interesting, methodologically sound, and well-written paper. 

Some minor comments and proposed modifications:

1. Lines 42-44: the sentence "Previous studies [...] information source" should be rephrased to provide more clarity. 

2. Line 86: the acronym STOPS is fully explained in the abstract, but in the main text, this is the first time encountered, so the complete form should be present here.

3. Lines 215-216. In section 6.3, there is reference to the limitations that derive from the sampling method but only as far as the sample size and the locality. If there are any other structural sociodemographic characteristics of the specific target group, these should also be reported (here or in 6.3)

4. Lines 317-324: Point 7) of section 6.1 is not clear how it is supported by the analysis reported by the Data Analysis section (5.1), in which there is no reference to the six propagation behaviours.

Reviewer 2 Report

The work presents a theme and an interest approach that can facilitate the implementation of future health programs. The structure and formal level of the work is worked with interest and adapts to the requirements and editorial line of the journal. Therefore, its publication without major changes is recommended.

Reviewer 3 Report

Research based on a rich source base. They are worth publishing because of their applied nature. It would be necessary to conduct research among representatives of other countries in order to make generalized conclusions. For this reason, it is recommended to modify the title of the article - by narrowing it down to the population of Chinese residents.
